# Effect of Temperature and Fermentation Time on Fermentation Characteristics and Biogenic Amine Formation of Oat Silage

Tingting Jia [1]  and Zhu Yu [2,*]

1   College of Animal Science and Technology, China Agricultural University, Beijing 100193, China; jiatt0829@cau.edu.cn
2   College of Grassland Science and Technology, China Agricultural University, Beijing 100193, China
*   Correspondence: 02059@cau.edu.cn

**Abstract:** Temperature is known to have a clear influence on the formation of biogenic amines during fermentation. To improve the quality of oat silage, the impact of ensiling temperature on the fermentation, microbiological and chemical characteristics, as well as biogenic amines (BAs) was investigated. Vacuum bag mini silos of oat forage were incubated at four different temperature levels (10, 20, 30 and 37 °C) and opened on day 0, 1, 3, 7, 15 and 60. All oat silages were sampled to evaluate the fermentation quality and biogenic amine production. Results showed that putrescine, cadaverine and tyramine were the most prevalent biogenic amines in oat silage, representing approximately about 90% of the total biogenic amines (TBAs) investigated. Ensiling increased the β–phenylethylamine, putrescine, cadaverine, histamine and tyramine accumulation in oat silage at the four incubation temperatures. On day 60, the β–phenylethylamine, cadaverine, histamine, tyramine and TBAs levels at a high temperature (37 °C) were significantly higher than those at a lower temperature (10, 20 and 30 °C); 10 °C fermentation increased the putrescine content in oat silage. A closed relationship between fermentation properties and BAs showed that the silages containing higher lactic acid, propionic acid and ammonia nitrogen and lower pH value had more BA content in oat silage. In conclusion, the ensiling process caused a significant increase in the amounts of BAs, except spermidine and spermine. The oat silage made in elevated temperature (30 and 37 °C) environments may accumulate more BAs than at a low temperature (10 °C), but low temperature (10 °C) fermentation may increase the putrescine levels in silage. The results suggested that ensiling at the proper temperature could retard BA formation and enhance the quality of oat silage.

**Keywords:** biogenic amines; ensiling temperatures; fermentation characteristics; oat silage

## 1. Introduction

Oat (*Avena sativa* L.) is an important forage cereal for ruminants in northeastern and northwestern China and are well adapted to a wide range of soil conditions and low temperatures [1]. Ensiling has been widely used in the processing of various forage crops to improve nutritional quality and extend the storage time. The quality of silage is produced through a wide range of biochemical and microbial changes. Though many methods to promote fermentation and improve the quality of silage have been widely explored, the formation of proteolytic products, namely, ammonia nitrogen and biogenic amines (BAs), is inevitable during ensiling, and these breakdown products might be strongly influenced by the ensiling temperature and time.

Many investigations have been conducted to determine the BA content in silage. Considerable amounts of BAs were shown to accumulate in silages [2–5]. The most significant BA formations in silage were of β–phenylethylamine, tryptamine, histamine, cadaverine, tyramine, putrescine, spermidine and spermine. The presence of low levels of BAs is not considered to be a risk; however, a large consumption of BAs may produce detrimental physiological effects in animals and humans [6,7]. Putrescine decreased nitrogen degradability in the rumen of steers, and tyramine increased the pH and isovalerate proportion in

rumen fluid [8]. The amines of putrescine, cadaverine, histamine, tyramine, spermine and spermidine are associated with reactions such as headaches, vomiting, nausea and diarrhea in humans [9–11]. The silage is an ideal environment for the formation of Bas, and the accumulation of BAs in silage is the response of microbial cells to acid stress; the loss of the carboxy group of BAs can increase the pH in and out of the cell [11,12]. Thus, knowledge of the levels of BAs in silage is necessary to assess the fermentation quality of silage.

The environmental temperature was one of the major factors shown to influence the acid accumulation and fermentation process during ensiling [13–15]. In general, a temperature of 20–30 °C is suitable for ensiling. Silage fermented at a lower temperature (10 °C) contained less lactic acid and ammonia, and higher pH values and residual water-soluble carbohydrates [14], while a high temperature (>37 °C) might cause clostridial fermentation, protein degradation and heat damage, thus resulting in poor silage quality [15,16]. Data have reported there to be more contents of BAs in fermented foods than silage. The temperature is an important factor influencing the formation of BAs in fermented food [9]. However, to date, scarce information is available about the effect of different temperatures on BAs' formation in silage.

The objective of this study was to evaluate the effects of ensiling temperatures on the levels of biogenic amines and the fermentation pattern of oat silage during the ensiling process. The effect of different ensiling temperatures (10, 20, 30 and 37 °C) on BA formation during fermentation, as well as the relationship between the content of BAs and basic fermentation properties (pH, lactic acid, acetic acid, propionic acid and ammonia nitrogen) were also studied.

## 2. Materials and Methods

### 2.1. Silage Materials and Ensiling

Oats (*Avena sativa* L.) were grown in Binhai New Area, Tianjin city, China (38°79′ N, 117°21′ E). The oats at heading stage (June 2017) were harvested with 4–5-cm stubble height after 80 days of growth. The oat materials were wilted on a polyethylene sheet for about 3 h; after wilting, the mean dry matter (DM) level of oat forage was 278.0 g/kg fresh matter (FM). Then, the oats' materials were chopped to a length of approximately 1–2 cm with a forage cutter and mixed uniformly. No additives or inoculants were added on the forage. The chopped oat forages were filled into polyethylene film bags (25 × 35 cm; 0.19-mm thickness), approximately 300 g of oat forage was collected in each polyethylene film bag. The air was removed by using a commercial vacuum sealer (FW3150; Fresh World Electric Co., Ltd., Guangzhou, China). After sealing, the vacuum bags were immediately transported to the laboratory and incubated at four different temperatures (10, 20, 30 and 37 °C) in an incubator (SPX-250, Beijing Luxi Tech. Co. Ltd., Beijing, China) in total darkness. All silos were opened after 0, 1, 3, 7, 15 and 60 days of ensiling. The experiment was designed with four storage temperatures (10, 20, 30 and 37 °C) × six ensiling durations (0, 1, 3, 7, 15 and 60 days). Each treatment included three repetitions, so a total of 72 bags were prepared.

### 2.2. Laboratory Analysis

Before ensiling, the harvested oat raw materials were uniformly mixed and sampled for laboratory analysis. The bags containing oat silage at the four different temperatures were opened, thoroughly mixed and randomly sampled after 1, 3, 7, 15 and 60 days of ensilage.

#### 2.2.1. Microbial Counts Analysis

The first portion of silage sample (20 g) was mixed with sterilized saline solution (180 mL) to analyze the counts of lactic acid bacteria, yeasts and molds. The counts of lactic acid bacteria were estimated on MRS agar plates, which were incubated for 2 days at 30 °C. Yeasts and molds were enumerated on yeast extract/peptone/dextrose agar and

salt Czapek Dox agar, respectively, after incubation at 28 °C for 3–5 days (all media were obtained from Beijing Aoboxing Biotech Co., Ltd. (Beijing, China)).

### 2.2.2. Fermentation Characteristics Analysis

The second portion of silage sample (20 g) was blended with 180 mL of distilled water and homogenized in a juicer for 2 min. Then, the mixture was filtered through four layers of cheesecloth and one layer of qualitative filter paper (15–20-μm pore size). The filtrate liquor was used to determine the pH and organic acid and ammonia nitrogen levels. Silage pH was determined by using a pH meter (PHS-3C, INESA Scientific Instrument Co., Ltd., Shanghai, China). The organic acid (lactic, acetic and propionic acids) levels were determined by HPLC (column: Shodex RS Pak KC-811, Showa Denko K.K., Kawasaki, Japan; detector: DAD, 210 nm, SPD-20A, Shimadzu Co., Ltd., Kyoto, Japan; eluent: 3 mmol/L $HClO_4$, at a flow rate of 1.0 mL/min; column temperature: 50 °C). The ammonia nitrogen ($NH_3$-N) concentration was analyzed by the phenol and sodium hypochlorite method [17].

### 2.2.3. Chemical Composition Analysis

The third portion of silage sample (100 g) was oven dried for 48 h at 65 °C to determine the dry matter (DM) content and chemical composition. Then oven-dried samples were milled with a vertical pulverizer, passed through a 1.0-mm screen and analyzed for chemical composition (water-soluble carbohydrate (WSC), crude protein (CP), neutral detergent fiber (NDF), acid detergent fiber (ADF) and hemicellulose (HC)). The WSC levels were measured by the anthrone method [18]. The CP content was determined by method of the Association of Official Analytical Chemists [19]. The NDF and ADF levels were measured according to the method of Van Soest et al. [20]. When quantifying the NDF content, heat-stable α-amylase and sodium sulfite were added. The hemicellulose content of each sample was estimated as the NDF content minus the ADF content.

### 2.2.4. Biogenic Amines Analysis

The fourth portion of each sample (50 g) was freeze dried in a vacuum freeze-drying machine for 3 days (FreeZone® 4.5 L, LABCONCO Corp., Kansas City, MO, USA) before determining the eight biogenic amines (tryptamine, β–phenylethylamine, putrescine, cadaverine, histamine, tyramine, spermidine and spermine). Lyophilized powder (2.5 g) was added to 10 mL of 50 g/L trichloroacetic acid and shaken on an orbital shaker for 30 min. The mixture was centrifuged for 10 min at $1800\times g$. The supernatant was filtered with filter paper. Trichloroacetic acid (50 g/L, 10 mL) was added to the remnant, and the whole mixture was shaken as described above. This mixture was centrifuged at $1800\times g$ for 10 min, and the supernatant was filtered. Finally, the volume of the filtrate was adjusted to 25 mL with 50 g/L trichloroacetic acid. One milliliter of the extract was placed in a 5-mL volumetric flask.

Then, sodium hydroxide (2 N, 200 μL), saturated sodium bicarbonate (300 μL) and dansyl chloride solution (10 mg/mL, 1 mL) were added to the sample extract. After incubation at 40 °C for 45 min in the dark, the mixture was treated with 100 μL of 25% ammonium hydroxide to remove the residual dansyl chloride. After 30 min at ambient temperature, the volume of the reaction mixture was adjusted to 5 mL with acetonitrile. Finally, the mixture was centrifuged for 5 min at $5000\times g$. The supernatant was filtered through a 0.22-μm syringe filter and subjected to HPLC (high performance liquid chromatography).

Separation was carried out on a C18 column (Reprosil-Pur Basic, 5 μm, 250 mm × 4.6 mm, Dr. Maisch GmbH) with a DAD. Gradient elution was performed with acetonitrile (solvent A) and 0.1 mol/L ammonium acetate (solvent B). The gradient program was run at a flow rate of 0.8 mL/min as follows: 50% A to 50% B in 0.01 min; 90% A to 10% B in 25 min; 90% A to 10% B in 35 min; 50% A to 50% B in 45 min. The column temperature was 30 °C, wavelength was 254 nm and injection volume of each sample was 20 μL.

### 2.3. Statistical Analysis

The data were analyzed by ANOVA by the general linear model univariate procedure of SPSS 19.0 software (SPSS Inc., Chicago, IL, USA). ANOVAs were performed with ensiling temperatures and ensiling durations as the two main parameters, and the interactions of these parameters were also examined. The mean values were compared using Duncan's multiple-range tests. Differences between means were considered significant when $p < 0.05$. Spearman correlation coefficient was used to analyze the relationship between biogenic amine content and pH, and ammonia nitrogen content of oat silage.

## 3. Results and Discussion

### 3.1. Initial Characteristics of Fresh Oat Forage

The chemical composition of oat forage before ensiling is presented in Table 1. The fresh oat material contained 278.0, 110.3, 129.5, 520.7, 274.7 and 246.0 g/kg DM, WSC, CP, aNDF, ADF and HC, respectively. The WSC content between 60 to 80 g/kg pre-ensiled forage is adequate for producing good quality forage. The WSC values for pre-ensiled oat forage (110.3 g/kg) are adequate for producing good quality silage.

**Table 1.** Chemical composition of oat forages before ensiling.

| Parameter | Mean Value | Standard Deviation |
|---|---|---|
| Dry matter (g/kg) | 278.0 | 16.14 |
| Neutral detergent fiber (g/kg DM) | 520.7 | 0.134 |
| Acid detergent fiber (g/kg DM) | 274.7 | 6.080 |
| Hemicellulose (g/kg DM) | 246.0 | 6.012 |
| Crude protein (g/kg DM) | 129.5 | 1.867 |
| Water-soluble carbohydrates (g/kg DM) | 110.3 | 3.232 |

DM: dry matter.

### 3.2. Fermentation Characteristics of Oat Silage

After 60 days of fermentation at 10, 20, 30 and 37 °C, the pH value of the oat forage decreased from 6.36 to 4.54, 4.18, 4.44 and 4.59, respectively, with the lactic acid concentration increasing from 0 to 33.0, 40.7, 44.4 and 29.6 g/kg DM, respectively (Table 2). At a low temperature (10 °C), the pH decreased slowly, and the final pH value was high (4.54), which contrasts with a study by Zhou et al. [14], who found that whole-plant corn silage reached low pH values (4.04) at 10 °C. This effect was probably due to the better ensilability of whole-plant corn than oat forage. Moreover, it was observed that a high temperature (37 °C) resulted in high pH values and increased ammonia nitrogen content. These findings indicate that high temperatures may enhance protein hydrolysis and reduce the fermentation quality of oat silage. Additionally, a significant increase in acetic acid concentrations was recorded when the oats were ensiled at a high temperature, suggesting that the oat silage at a high temperature may be affected by the heterofermentative pathway [14]. The pH value was lower and the lactic acid content was higher when oat ensiling was carried out at moderate temperatures (20 and 30 °C) compared with low or high temperatures (10 or 37 °C), which indicates that the fermentation quality at moderate temperatures (20 and 30 °C) was better than that at low or high temperatures. Moreover, the acetic acid and ammonia nitrogen contents at 37 °C were significantly ($p < 0.05$) higher than those at low temperatures on day 60; the ammonia nitrogen content increased as the temperature increased and ensiling progressed. These results indicate that the increased ensiling temperature can stimulate forage protein degradation.

**Table 2.** Changes in fermentation properties during ensilage of oat forage.

| Parameter | Days of Ensiling | | | | | | SEM | p-Value | | |
|---|---|---|---|---|---|---|---|---|---|---|
| | 0 | 1 | 3 | 7 | 15 | 60 | | T | D | T × D |
| pH value | | | | | | | | | | |
| 10 °C | 6.36 | 6.24 | 6.27 [a] | 6.02 [a] | 5.28 [a] | 4.54 [a] | 0.027 | <0.001 | <0.001 | <0.001 |
| 20 °C | 6.36 | 6.02 | 5.74 [bc] | 5.28 [b] | 4.61 [b] | 4.18 [b] | | | | |
| 30 °C | 6.36 | 6.05 | 5.97 [ab] | 5.28 [b] | 4.97 [a] | 4.44 [a] | | | | |
| 37 °C | 6.36 | 5.95 | 5.26 [c] | 4.61 [c] | 4.62 [b] | 4.59 [a] | | | | |
| Lactic acid (g/kg DM) | | | | | | | | | | |
| 10 °C | ND | ND | ND [b] | ND [c] | 3.2 [b] | 33.0 [c] | 0.295 | <0.001 | <0.001 | <0.001 |
| 20 °C | ND | ND | ND [b] | 6.3 [b] | 21.4 [a] | 40.7 [b] | | | | |
| 30 °C | ND | ND | ND [b] | 6.6 [b] | 22.4 [a] | 44.4 [a] | | | | |
| 37 °C | ND | ND | 8.6 [a] | 17.8 [a] | 27.8 [a] | 29.6 [d] | | | | |
| Acetic acid (g/kg DM) | | | | | | | | | | |
| 10 °C | ND | 12.7 [a] | 12.1 [a] | 12.7 | 11.7 [ab] | 11.2 [b] | 0.162 | 0.001 | <0.001 | <0.001 |
| 20 °C | ND | 13.0 [b] | 9.5 [b] | 9.4 | 11.6 [ab] | 13.8 [b] | | | | |
| 30 °C | ND | 14.3 [b] | 8.3 [b] | 8.8 | 10.2 [a] | 22.9 [a] | | | | |
| 37 °C | ND | 11.6 [b] | 8.6 [b] | 8.1 | 11.9 [b] | 26.2 [a] | | | | |
| Propionic acid (g/kg DM) | | | | | | | | | | |
| 10 °C | ND | ND | ND | 0.22 [b] | 0.22 [c] | 0.25 | 0.002 | <0.001 | <0.001 | <0.001 |
| 20 °C | ND | ND | 0.25 | 0.29 [a] | 0.29 [a] | 0.27 | | | | |
| 30 °C | ND | ND | 0.25 | 0.28 [a] | 0.28 [ab] | 0.27 | | | | |
| 37 °C | ND | ND | 0.26 | 0.25 [a] | 0.25 [bc] | 0.28 | | | | |
| Ammonia nitrogen (g/kg TN) | | | | | | | | | | |
| 10 °C | 2.3 | 7.9 [d] | 12.0 [c] | 17.8 [c] | 18.9 [d] | 20.1 [b] | 0.696 | <0.001 | <0.001 | <0.001 |
| 20 °C | 2.3 | 12.0 [bc] | 19.1 [b] | 25.2 [b] | 31.3 [c] | 40.0 [b] | | | | |
| 30 °C | 2.3 | 16.5 [b] | 26.8 [a] | 35.0 [a] | 54.2 [b] | 72.2 [a] | | | | |
| 37 °C | 2.3 | 21.6 [a] | 29.3 [a] | 38.5 [a] | 63.5 [a] | 76.8 [a] | | | | |

DM: dry matter; TN: total nitrogen. ND: not detected. T: ensiling temperatures; D: ensiling days; T × D: interaction between ensiling temperatures and days. Means within the same column with difference superscripts (a–d) differ significantly from each other ($p < 0.05$).

### 3.3. Microbial Counts and Chemical Composition of Oat Silage

The microbial counts of the fresh oat material were 7.51, 7.50 and 4.70 log cfu/g fresh matter (FM) for lactic acid bacteria, yeasts and molds, respectively (Table 3). The counts of lactic acid bacteria at 10, 20 and 30 °C were higher than those at 37 °C on days 7, 15 and 60. The yeasts survived 60 days at a low temperature (10 °C), whereas these organisms were undetectable (<2 log cfu/g FM) on day 15 at 20 °C and on day 7 at 30 and 37 °C. The counts of molds were significantly ($p < 0.05$) higher at a low temperature (10 °C) than those at the other incubation temperatures. In addition, the mold counts were under the detection level (<2 log cfu/g FM) on day 3 at 20, 30 and 37 °C. After 60 days of fermentation at a low temperature, undesirable microorganisms were seen to survive due to the low rate of decrease in the pH and low levels of lactic acid [14].

**Table 3.** Changes in microbial counts during ensilage of oat forage.

| Parameter | Days of Ensiling | | | | | | SEM | p-Value | | |
|---|---|---|---|---|---|---|---|---|---|---|
| | 0 | 1 | 3 | 7 | 15 | 60 | | T | D | T × D |
| LAB (log cfu/g FM) | | | | | | | | | | |
| 10 °C | 7.51 | 7.62 [b] | 8.03 | 8.18 [a] | 8.16 | 8.12 [a] | 0.048 | 0.022 | 0.036 | 0.004 |
| 20 °C | 7.51 | 8.16 [ab] | 8.16 | 8.19 [a] | 8.15 | 8.18 [a] | | | | |
| 30 °C | 7.51 | 8.91 [a] | 8.15 | 8.02 [a] | 7.79 | 7.88 [a] | | | | |
| 37 °C | 7.51 | 8.34 [ab] | 8.20 | 7.73 [b] | 7.65 | 6.96 [b] | | | | |
| Yeasts (log cfu/g FM) | | | | | | | | | | |
| 10 °C | 7.50 | 7.40 [a] | 6.77 [a] | 6.22 [a] | 3.64 [a] | 2.30 [a] | 0.029 | <0.001 | <0.001 | <0.001 |
| 20 °C | 7.50 | 7.33 [a] | 5.65 [b] | 2.72 [b] | <2 [b] | <2 [b] | | | | |
| 30 °C | 7.50 | 7.30 [a] | 4.16 [c] | <2 [c] | <2 [b] | <2 [b] | | | | |
| 37 °C | 7.50 | 6.74 [b] | <2 [d] | <2 [c] | <2 [b] | <2 [b] | | | | |
| Molds (log cfu/g FM) | | | | | | | | | | |
| 10 °C | 4.70 | 3.80 [a] | 2.80 [a] | 2.50 [a] | 2.15 [a] | 2.30 [a] | 0.023 | <0.001 | <0.001 | <0.001 |
| 20 °C | 4.70 | 3.17 [b] | <2 [b] | <2 [b] | <2 [b] | <2 [b] | | | | |
| 30 °C | 4.70 | 2.77 [bc] | <2 [b] | <2 [b] | <2 [b] | <2 [b] | | | | |
| 37 °C | 4.70 | 2.45 [c] | <2 [b] | <2 [b] | <2 [b] | <2 [b] | | | | |

LAB: lactic acid bacteria; log: denary logarithm of the numbers; cfu: colony-forming units; FM: fresh matter. T: ensiling temperatures; D: ensiling days; T × D: interaction between ensiling temperatures and days. Means within the same column with difference superscripts (a–c) differ significantly from each other ($p < 0.05$).

Except for WSC, the final nutrient (CP, NDF, ADF and HC) concentrations were not significant ($p > 0.05$) among the four temperatures on day 60. The WSC content at 10 °C was consistently relatively higher than that at 20, 30 and 37 °C and decreased with increasing ensiling temperatures (Table 4). Ensiling temperature significantly influenced the fermentation characteristics but did not modify the final nutrient (CP, NDF, ADF and HC) concentrations, except for WSC. The WSC content in the oat forage decreased significantly during fermentation, which resulted from the growth of lactic acid bacteria that use sugar as a substrate.

**Table 4.** Chemical composition of oat silage at different temperature after 60 d of ensiling.

| Parameter (g/kg DM) | Temperature Treatment | | | | SEM | p-Value |
|---|---|---|---|---|---|---|
| | 10 | 20 | 30 | 37 | | |
| Dry Matter (g/kg) | 270.3 | 270.9 | 269.8 | 271.0 | 3.286 | 0.999 |
| Water-soluble carbohydrate (g/kg DM) | 25.3 [a] | 6.37 [b] | 4.21 [bc] | 2.83 [c] | 2.601 | <0.001 |
| Crude protein (g/kg DM) | 129.0 | 127.6 | 125.8 | 123.6 | 0.833 | 0.057 |
| Neutral detergent fiber (g/kg DM) | 490.4 | 511.6 | 529.0 | 531.4 | 7.678 | 0.289 |
| Acid detergent fiber (g/kg DM) | 259.7 | 277.5 | 293.9 | 295.8 | 6.137 | 0.144 |
| Hemicellulose (g/kg DM) | 230.7 | 234.1 | 235.1 | 235.6 | 2.908 | 0.967 |

DM: dry matter. Means within the same line with difference superscripts (a–c) differ significantly from each other ($p < 0.05$).

*3.4. Biogenic Amines' Content of Oat Silage*

High concentrations of biogenic amines have been found in silages made from proteinaceous forage [3,5], while large amounts of biogenic amines were also observed in silages prepared from low protein forage, such as maize silage [21]. In the present study, eight BAs were determined in oat silages made in different temperature environments. The putrescine, cadaverine and tyramine levels increased significantly ($p < 0.001$) as ensiling progressed; these three amines were the predominant BAs detected in oat silage and comprised approximately 90% of the total biogenic amines (Table 5). In agreement with our findings, Selwet et al. [22] showed that putrescine, cadaverine and tyramine are the main biogenic amines in alfalfa silage. Tryptamine and spermine were detected at a low level in maize silage [21]; however, these two amines were not detected in the present study throughout the fermentation.

**Table 5.** Changes in biogenic amines' content (mg/kg dry matter) during ensilage of oat forage.

| Parameter | Days of Ensiling | | | | | | SEM | p-Value | | |
|---|---|---|---|---|---|---|---|---|---|---|
| | 0 | 1 | 3 | 7 | 15 | 60 | | T | D | T × D |
| PHE | | | | | | | | | | |
| 10 °C | ND | ND | ND [b] | ND [b] | ND [c] | ND [c] | 0.513 | <0.001 | <0.001 | <0.001 |
| 20 °C | ND | ND | ND [b] | ND [b] | ND [c] | ND [c] | | | | |
| 30 °C | ND | ND | ND [b] | 3.5 [b] | 19.0 [b] | 59.7 [b] | | | | |
| 37 °C | ND | ND | 25.5 [a] | 29.9 [a] | 53.9 [a] | 92.4 [a] | | | | |
| PUT | | | | | | | | | | |
| 10 °C | 36.0 | 56.5 [c] | 105.1 [b] | 179.4 [a] | 212.1 [a] | 289.8 | 3.322 | <0.001 | <0.001 | 0.001 |
| 20 °C | 36.0 | 89.4 [a] | 120.9 [ab] | 135.4 [b] | 156.4 [b] | 258.1 | | | | |
| 30 °C | 36.0 | 100.2 [a] | 131.2 [a] | 142.5 [ab] | 161.7 [b] | 288.4 | | | | |
| 37 °C | 36.0 | 69.9 [b] | 77.0 [c] | 93.4 [c] | 98.5 [c] | 305.1 | | | | |
| CAD | | | | | | | | | | |
| 10 °C | ND | ND [d] | 13.4 [c] | 72.4 [c] | 156.6 [b] | 135.9 [c] | 9.738 | <0.001 | <0.001 | <0.001 |
| 20 °C | ND | 72.6 [c] | 197.3 [b] | 202.2 [b] | 202.9 [b] | 264.9 [bc] | | | | |
| 30 °C | ND | 125.2 [b] | 260.8 [a] | 294.0 [a] | 315.3 [a] | 448.8 [b] | | | | |
| 37 °C | ND | 172.9 [a] | 286.0 [a] | 297.6 [a] | 376.2 [a] | 1056.2 [a] | | | | |
| HIS | | | | | | | | | | |
| 10 °C | ND | ND | ND | ND | ND | ND [b] | 1.066 | <0.001 | <0.001 | <0.001 |
| 20 °C | ND | ND | ND | ND | ND | ND [b] | | | | |
| 30 °C | ND | ND | ND | ND | ND | 21.5 [b] | | | | |
| 37 °C | ND | ND | ND | ND | ND | 106.1 [a] | | | | |
| TYR | | | | | | | | | | |
| 10 °C | ND | ND [b] | ND [d] | 12.5 [d] | 70.2 [d] | 292.6 [c] | 4.185 | <0.001 | <0.001 | <0.001 |
| 20 °C | ND | ND [b] | 61.2 [c] | 136.6 [c] | 205.6 [c] | 521.2 [b] | | | | |
| 30 °C | ND | 4.0 [b] | 189.9 [b] | 366.7 [b] | 486.3 [b] | 720.6 [a] | | | | |
| 37 °C | ND | 154.7 [a] | 473.0 [a] | 672.3 [a] | 739.6 [a] | 775.7 [a] | | | | |
| SPD | | | | | | | | | | |
| 10 °C | 3.3 | 3.3 | 2.3 [b] | 2.7 [b] | 3.3 [b] | ND | 0.177 | <0.001 | <0.001 | <0.001 |
| 20 °C | 3.3 | 5.3 | 4.7 [ab] | 6.6 [a] | 7.3 [a] | ND | | | | |
| 30 °C | 3.3 | 4.7 | 3.7 [b] | 5.0 [ab] | 3.3 [b] | ND | | | | |
| 37 °C | 3.3 | 3.0 | 7.7 [a] | 7.6 [a] | ND [c] | ND | | | | |
| TBA | | | | | | | | | | |
| 10 °C | 39.3 | 59.9 [d] | 120.8 [d] | 266.9 [d] | 442.2 [d] | 718.3 [d] | 10.0 | <0.001 | <0.001 | <0.001 |
| 20 °C | 39.3 | 167.3 [c] | 384.1 [c] | 480.8 [c] | 572.1 [c] | 1044.2 [c] | | | | |
| 30 °C | 39.3 | 234.1 [b] | 585.5 [b] | 811.6 [b] | 985.7 [b] | 1539.0 [b] | | | | |
| 37 °C | 39.3 | 400.4 [a] | 869.2 [a] | 1100.9 [a] | 1268.2 [a] | 2335.6 [a] | | | | |

TRP: tryptamine; PHE: β–phenylethylamine; PUT: putrescine; CAD: cadaverine; HIS: histamine; TYR: tyramine; SPD: spermidine; SPM: spermine; TBA: total biogenic amines. ND: not detected. T: ensiling temperatures; D: ensiling days; T × D: interaction between ensiling temperatures and days. Means within the same column with difference superscripts (a–d) differ significantly from each other ($p < 0.05$).

The putrescine contents in final oat silage ranged from 258.1 mg/kg to 305.1 mg/kg, which was higher than the value reported by Steidlová and Kalač [23] for untreated false oat silage (182 mg/kg) after 4 months storage at 22 °C. Oat silages fermented at four temperatures contained approximately the same quantities of putrescine on the final day 60; the high content of putrescine in low temperature (10 °C)-fermented silages may be caused by the metabolism of psychrophilic pseudomonas [24]. The cadaverine contents found in final oat silage (ranging from 135.9 mg/kg to 1056.2 mg/kg) were higher than those reported by Steidlová and Kalač [25] for maize silage stored at 22 °C for 4 months. At the end of the ensiling, tyramine levels ranged between 292.6 and 775.7 mg/kg in oat silage, which was higher than the value reported by Scherer et al. [5] for untreated lucerne silage (150 mg/kg) after 120 days of ensiling. The maximum tyramine level (775.7 mg/kg) was found in 37 °C fermented oat silage. β–Phenylethylamine was only detected in the later stage of fermentation, increased from ND to a level of 59.7 mg/kg (30 °C) and 92.4 mg/kg (37 °C) at the final day of fermentation. Concentrations of spermidine were detected at a

low level in the first 15 days of fermentation; spermine contents were below the detection limit throughout the ensiling process. Spermidine and spermine are generated from the decomposition of methionine but are also produced from putrescine [26]. Tryptamine was not detected during ensilage of the oat material at four ensiling temperatures. These results suggest that the kinds and amounts of BAs may differ widely among different grass silages and that differences could be attributed to variations in silage making and grass material.

In the present study, at the end of ensiling, the numbers of lactic acid bacteria, yeasts and molds decreased, while the BAs' content increased greatly. pH values influence bacterial growth and also the amino acid decarboxylase activity of bacteria. The activity of amino acid decarboxylase is stronger in acidic environments, with optimal values within the pH range of 4.0–5.5 [27]. Decarboxylation is a mechanism for cells to resist acid stress; the decarboxylase pathway can carry out primary metabolism under critical environmental conditions [28]. For microorganisms lacking respiratory chains, such as most lactic acid bacteria, decarboxylase activity is usually expressed independently from cell viability, and the decarboxylase can remain active in harsh environments even after cell lysis [29,30]. Therefore, although the low pH of oat silage can influence the activity of microorganisms, it may also promote the activity of amino acid decarboxylase and thus facilitate the production of biogenic amines.

This study found more intensive liberation of BAs (β–phenylethylamine, cadaverine, histamine, tyramine and total biogenic amines) and ammonia nitrogen content in the high (37 °C) temperature-fermented silages (Table 6). The high levels of BAs and ammonia nitrogen in the high (37 °C) temperature-fermented silages may be a result of enhanced decarboxylation and deamination of protein. In general, except for putrescine, BAs' production was increased by the increasing of the storage temperature. This implies that more BAs may be accumulated in silage that is stored in a high temperature environment.

### 3.5. Relationship between Biogenic Amines and Fermentation Properties

The Pearson relationship was used to define whether there existed any relationship between the BAs and the fermentation characteristics. In general, significant negative correlations ($p < 0.01$) existed between the TBA contents and also some biogenic amines (putrescine, cadaverine and tyramine) and the pH value of silage at all ensiling temperatures (Table 6). The pH has a great influence on the activity of amino acid decarboxylase, which is stronger in a lower pH environment [27]. Decarboxylation is a defense mechanism of cells against acid stress; the loss of carboxyl groups increases the intracellular and extracellular pH values [31]. Significant positive correlations ($p < 0.05$) were noted between the aforementioned amines and lactic acid, propionic acid and ammonia nitrogen at all ensiling temperatures. The acidic environment caused by the accumulation of organic acid in oat silage may have increased the production of BAs. Van Os et al. [2] and Nishino et al. [32] found significantly positive correlations between ammonia nitrogen and BAs [2,32]. This can also be supported by the current data. Although the ammonia nitrogen and biogenic amines arise from two different reactions, deamination and decarboxylation, the close correlation between ammonia nitrogen and BAs indicates that both reactions are concomitant processes of amino acid degradation during ensiling. Biogenic amines and ammonia nitrogen tend to increase silage pH in order to buffer the acidifying effect of acids.

**Table 6.** Correlation analysis (Pearson coefficient) between biogenic amines and fermentation properties of samples.

| Items | Biogenic Amines | | | | | | | | |
|---|---|---|---|---|---|---|---|---|---|
| | TRP | PHE | PUT | CAD | HIS | TYR | SPD | SPM | TBA |
| **Ensiling at 10 °C** | | | | | | | | | |
| pH | NA | NA | −0.885 ** | −0.817 ** | NA | −0.932 ** | 0.389 | NA | −0.961 ** |
| Lactic acid | NA | NA | 0.778 ** | 0.536 * | NA | 0.985 ** | −0.581 * | NA | 0.871 ** |
| Acetic acid | NA | NA | −0.516 * | −0.582 * | NA | −0.552 * | −0.075 | NA | −0.596 * |
| Propionic acid | NA | NA | 0.831 ** | 0.826 ** | NA | 0.828 ** | −0.333 | NA | 0.897 ** |
| Ammonia nitrogen | NA | NA | 0.890 ** | 0.860 ** | NA | 0.617 * | −0.258 | NA | 0.828 ** |
| **Ensiling at 20 °C** | | | | | | | | | |
| pH | NA | NA | −0.838 ** | −0.761 ** | NA | −0.881 ** | 0.297 | NA | −0.891 ** |
| Lactic acid | NA | NA | 0.944 ** | 0.663 ** | NA | 0.965 ** | −0.576 ** | NA | 0.938 ** |
| Acetic acid | NA | NA | 0.432 | −0.141 | NA | 0.398 | −0.409 | NA | 0.296 |
| Propionic acid | NA | NA | 0.673 ** | 0.613 * | NA | 0.695 ** | −0.059 | NA | 0.710 ** |
| Ammonia nitrogen | NA | NA | 0.831 ** | 0.733 ** | NA | 0.814 ** | −0.410 | NA | 0.840 ** |
| **Ensiling at 30 °C** | | | | | | | | | |
| pH | NA | −0.867 ** | −0.861 ** | −0.830 ** | −0.724 ** | −0.965 ** | 0.632 * | NA | −0.950 ** |
| Lactic acid | NA | 0.943 ** | 0.914 ** | 0.840 ** | 0.842 ** | 0.906 ** | −0.787 ** | NA | 0.933 ** |
| Acetic acid | NA | 0.829 ** | 0.776 ** | 0.467 | 0.899 ** | 0.512 | −0.725 ** | NA | 0.588 * |
| Propionic acid | NA | 0.610 * | 0.618 ** | 0.719 ** | 0.406 | 0.862 ** | −0.368 | NA | 0.807 ** |
| Ammonia nitrogen | NA | 0.882 ** | 0.882 ** | 0.909 ** | 0.714 ** | 0.945 ** | −0.729 ** | NA | 0.963 ** |
| **Ensiling at 37 °C** | | | | | | | | | |
| pH | NA | −0.740 ** | −0.427 | −0.479 | −0.330 | −0.956 ** | 0.175 | NA | −0.708 ** |
| Lactic acid | NA | 0.894 ** | 0.603 * | 0.656 ** | 0.530 * | 0.932 ** | −0.523 * | NA | 0.836 ** |
| Acetic acid | NA | 0.763 ** | 0.797 ** | 0.906 ** | 0.878 ** | 0.392 | −0.681 ** | NA | 0.818 ** |
| Propionic acid | NA | 0.743 ** | 0.526 * | 0.521 * | 0.424 | 0.880 ** | −0.396 | NA | 0.723 ** |
| Ammonia nitrogen | NA | 0.919 ** | 0.740 ** | 0.827 ** | 0.752 ** | 0.798 ** | −0.682 ** | NA | 0.912 ** |

TRP: tryptamine; PHE: β–phenylethylamine; PUT: putrescine; CAD: cadaverine; HIS: histamine; TYR: tyramine; SPD: spermidine; SPM: spermine; TBA: total biogenic amines; NA, not available; * correlation is significant at the 0.05 level; ** correlation is significant at the 0.01 level.

## 4. Conclusions

The ensiling process caused a significant increase in BAs' content. The β–phenylethylamine, cadaverine, histamine, tyramine and total biogenic amines increased with the elevated temperature during the ensilage of oat silage. Low temperature fermentation increased the putrescine content in oat silage. A closed relationship between the fermentation properties and BAs showed that the silages containing higher lactic acid, propionic acid and ammonia nitrogen, and a lower pH value, had more BAs' content in oat silage.

**Author Contributions:** Conceptualization, T.J. and Z.Y.; methodology and validation, T.J.; formal analysis, T.J. and Z.Y.; investigation, T.J.; resources, T.J.; data curation, T.J.; writing—original draft preparation, T.J. and Z.Y.; writing—review and editing, T.J. and Z.Y.; visualization and supervision, Z.Y.; project administration, Z.Y.; funding acquisition, Z.Y. All authors have read and agreed to the published version of the manuscript.

**Funding:** The work was financially supported by China Agriculture Research System of MOF and MARA; Demonstration Project of Exploitation and Utilization of High-Quality Green and Rough Feed Resources, grant number 16190051.

**Conflicts of Interest:** The authors declare no conflict of interest.

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
