# Peer review of "Effect of Temperature and Fermentation Time on Fermentation Characteristics and Biogenic Amine Formation of Oat Silage"

_fermentation, doi:10.3390/fermentation8080352_

Round 1
Reviewer 1 Report
95-99: Does it make sense to include the fate of the four portions of samples here if it is described in the next four subsections?
135: It is not clear what exactly a 'table concentrator' is.
Table 2: It is included in subsection which concerns initial characteristics of oat substrate and is formed by pooling the data from all the days of the fermentation process. Does this make sense given the scope of the subsection? Furthermore, it seems to be a repetition of data shown later in Table 6 (however, the data do not match completely). In my opinion, Table 2 should be removed from the MS.
Table 5: For clarity, the title of the table should include the time point in the fermentation in which the values were determined.
An interesting observation that the Authors seem to disregard completely can be made from the kinetic data on LAB counts and BAs content at different days of the process. There is a huge increase of biogenic amines between the last time points of the fermentation which coincides with decreasing counts of LAB. In other words microbial activity decreases in this period wile the generation of BAs continues at a high rate. What could be the reason behind the observation? This should be discussed.
Author Response
Response to Reviewer1 Comments
Dear reviewer, thanks a lot for your review of our manuscript. Now we are submitting the revised version after incorporating the comments of the reviewers and making the improvements. We have studied the comments carefully and made the correction. Revised portions are marked with tracked changes to highlight the revisions.
(1) 95-99: Does it make sense to include the fate of the four portions of samples here if it is described in the next four subsections?
Response: Thank you for suggestion. We have deleted the description (95-99) of four portions of samples in the MS.
(2) 135: It is not clear what exactly a 'table concentrator' is.
Response: We are sorry for the confusion. we have modified “in a table concentrator” with “on an orbital shaker”.
(3) Table 2: It is included in subsection which concerns initial characteristics of oat substrate and is formed by pooling the data from all the days of the fermentation process. Does this make sense given the scope of the subsection? Furthermore, it seems to be a repetition of data shown later in Table 6 (however, the data do not match completely). In my opinion, Table 2 should be removed from the MS.
Response: We have removed the Table 2 in the MS. The corresponding discussion of biogenic amines was also modified.
(4) Table 5: For clarity, the title of the table should include the time point in the fermentation in which the values were determined.
Response: We have added the time point in the title of Table 4.
(5) An interesting observation that the Authors seem to disregard completely can be made from the kinetic data on LAB counts and BAs content at different days of the process. There is a huge increase of biogenic amines between the last time points of the fermentation which coincides with decreasing counts of LAB. In other words microbial activity decreases in this period wile the generation of BAs continues at a high rate. What could be the reason behind the observation? This should be discussed.
Response: Thanks to reviewers for your constructive comments. We have added the reason in the discussion. In the present study, at the end of ensiling, the numbers of lactic acid bacteria, yeasts and molds decreased, while the BAs content increased greatly. The pH influence bacterial growth and also the amino acid decarboxylase activity of bacteria. The activity of amino acid decarboxylase is stronger in acidic environments, with optimal values within the pH range of 4.0-5.5. The decarboxylation is a mechanism for cells to resist acid stress, the decarboxylase pathway can carry out primary metabolism under critical environmental conditions. For microorganisms lacking respiratory chains, such as most lactic acid bacteria, decarboxylase activity is usually expressed independently from cell viability, and the decarboxylase can remain active in harsh environment even after cell lysis. Therefore, although the low pH of oat silage can influence the activity of microorganisms, it may also promote the activity of amino acid decarboxylase and thus facilitate the production of biogenic amines.

Reviewer 2 Report
The thematic scope of the manuscript submitted for review "Effect of temperature and fermentation time on fermentation characteristics and biogenic amines formation of oat silage" lies within the scope of the journal FERMENTATION.
The research concerns the determination of the influence of the fermentation temperature in the production of oat silage on the formation of biogenic amines and the quality characteristics of this silage. In addition, the authors determined the relationship between the fermentation temperature, the content of biogenic mines and the characteristics of the pH type, the content of lactic and propionic acids, the content of various forms of nitrogen, etc. I have no comments on the design of the experiment or the selection of research methods. The abstract (lines 27-29) and the introduction (lines 46-54) require some changes.
Throughout the manuscript, Authors should introduce the abbreviations they propose (ie, for example, the abbreviation "biogenic amines" - BAs). Minor deficiencies in the way of citation (e.g. in line 174, 196, 252 etc.). All comments, the larger and the smaller ones, were marked in the sent pdf file in the review mode. After taking these amendments into account, in my opinion, the article can be referred for further editorial work.

Author Response
Response to Reviewer2 Comments
Dear reviewer, thanks a lot for your review of our manuscript. Now we are submitting the revised version after incorporating the comments of the reviewers and making the improvements. We have studied the comments carefully and made the correction. Revised portions are marked with tracked changes to highlight the revisions.
(1) The thematic scope of the manuscript submitted for review "Effect of temperature and fermentation time on fermentation characteristics and biogenic amines formation of oat silage" lies within the scope of the journal FERMENTATION. The research concerns the determination of the influence of the fermentation temperature in the production of oat silage on the formation of biogenic amines and the quality characteristics of this silage. In addition, the authors determined the relationship between the fermentation temperature, the content of biogenic mines and the characteristics of the pH type, the content of lactic and propionic acids, the content of various forms of nitrogen, etc. I have no comments on the design of the experiment or the selection of research methods. The abstract (lines 27-29) and the introduction (lines 46-54) require some changes.
Response: We have modified the abstract (lines 27-29) and the introduction (lines 46-54) according to your suggestion.
(2) Throughout the manuscript, Authors should introduce the abbreviations they propose (ie, for example, the abbreviation "biogenic amines" - BAs). Minor deficiencies in the way of citation (e.g. in line 174, 196, 252 etc.). All comments, the larger and the smaller ones, were marked in the sent pdf file in the review mode. After taking these amendments into account, in my opinion, the article can be referred for further editorial work.
Response: We have introduced the abbreviations “BAs” in the MS and modified the MS according to the opinion marked in the pdf file.

Round 2
Reviewer 1 Report
While the manuscript was vastly improved and the quality of content does not rise concern, there is still need for a thorough edition of language in terms of style.